# Coccidioidomycosis (Valley Fever), Soil Moisture, and El Nino Southern Oscillation in California and Arizona

**DOI:** 10.3390/ijerph19127262

**Published:** 2022-06-14

**Authors:** Kenneth J. Tobin, Sugam Pokharel, Marvin E. Bennett

**Affiliations:** Center for Earth and Environmental Studies, Texas A&M International University, Laredo, TX 78041, USA; sugampokharel@dusty.tamiu.edu (S.P.); mbennett@tamiu.edu (M.E.B.)

**Keywords:** coccidioidomycosis, Valley fever, ENSO, ESA-CCI, SMERGE, soil moisture

## Abstract

The soil-borne fungal disease coccidioidomycosis (Valley fever) is prevalent across the southwestern United States (US). Previous studies have suggested that the occurrence of this infection is associated with anomalously wet or dry soil moisture states described by the “grow and blow” hypothesis. The growth of coccidioidomycosis is favored by moist conditions both at the surface and in the root zone. A statistical analysis identified two areas in Arizona and central California, with a moderate-to-high number of coccidioidomycosis cases. A Wavelet Transform Coherence (WTC) analysis between El Nino Southern Oscillation (ENSO), coccidioidomycosis cases, surface soil moisture (SSM; 0 to 5 cm) from European Space Agency-Climate Change Initiative (ESA-CCI), and shallow root zone soil moisture (RZSM; 0 to 40 cm depth) from Soil MERGE (SMERGE) was executed for twenty-four CA and AZ counties. In AZ, only SSM was modulated by ENSO. When case values were adjusted for overreporting between 2009 to 2012, a moderate but significant connection between ENSO and cases was observed at a short periodicity (2.1 years). In central CA, SSM, RZSM, and cases all had a significant link to ENSO at longer periodicities (5-to-7 years). This study provides an example of how oceanic-atmospheric teleconnections can impact human health.

## 1. Introduction

Coccidioidomycosis (or Valley fever) is a soil-borne fungal disease caused by the fungus *Coccidioides*. This affliction is common in arid regions of the southwestern United States, Mexico, and Central and South America. Inhalation of microscopic fungal spores causes flu-like symptoms in approximately 40% of the infections that are symptomatic [1] and of those nearly half will require hospitalization. Valley fever can affect people of all ages, with protracted illness that can last weeks to months. However, this affliction is most common in the elderly [2]. Rates of Valley fever infections have increased since the 1990’s [3,4]. With projected climate change during this century, hotter and drier conditions will likely cause an eastward expansion of coccidioidomycosis from its current endemic range within the southwestern United States (California, Arizona) into the US Great Plains [5]. By the end of the 21st century, Valley fever cases are projected to increase by 50%, forming an acute burden on the public health system.

A commonly accepted hypothesis to explain the occurrence of this disease is referred to as “grow and blow,” where moist soil conditions support the development of *Coccidioides* spores followed by drier conditions that allow for spore dispersion that leads to the onset of the disease in humans [1]. Therefore, environmental factors that affect landscape conditions have a strong influence on coccidioidomycosis occurrence. Specifically, climate factors such as temperature, wind speed, and precipitation are relevant and have been previously examined [6,7]. In addition, the state of the land surface can be modulated by anthropogenic influences such as agricultural activity, brush clearing, and urban development that can increase dust emissions, thereby providing a means for *Coccidioides* spore transportation [8]. Although Comrie [9] showed no consistent connection between dust storms and Valley fever. However, soil moisture remains an important environmental parameter, that controls both fungal growth and dispersion [10].

Previous efforts focusing on linking coccidioidomycosis with soil moisture have been based on indirect proxies such as Normalized Vegetation Difference Index (NVDI) [11]. Other studies were tied to sparse in situ soil moisture records [12] that were leveraged with surface satellite soil moisture retrievals from the Advanced Microwave Scanning Radiometer-EOS (AMSR-E) [13] and simple, lumped bucket modeling [14]. Two other soil moisture datasets that can be used in this situation include surface soil moisture (0 to 5 cm depth; SSM) estimates from the European Space Agency Climate Change Initiative (ESA-CCI) [15] and root zone soil moisture (0 to 40 cm depth; RZSM) from Soil MERGE (SMERGE) [16]. These products have the principal advantage of providing a multi-decadal daily record (1979 to 2019) facilitating long-term analysis. In addition, SMERGE is based, in part, on the complex and widely utilized Noah land surface model.

In the southwestern continental United States (CONUS), a major source of interannual variability in climatic conditions results from the El Nino Southern Oscillation (ENSO). However, there is a distinct difference in the hydroclimatic regime in Arizona (AZ) and California (CA) which impacts how ENSO affects soil moisture in these states. In AZ, the majority of precipitation is derived from thunderstorms associated with the summer North American Monsoon (July to September). However, AZ precipitation is more impacted by ENSO in the spring and fall [17]. Conversely, in CA, the rainy season is between October and April and precipitation is derived from synoptic-scale, mid-latitude cyclones that can tap into atmospheric rivers of moisture from the Pacific Basin (e.g., Pineapple Express). Many prior studies [18,19,20,21] have linked positive or negative anomalies in precipitation with periods of El Nino or La Nina, respectively. The goal of this study is to examine whether ENSO is a causality factor in understanding the interannual variability of both soil moisture and coccidioidomycosis cases in AZ and CA. Unlike prior studies, our work spans nearly two decades (2001 to 2018) with temporally and spatially continuous soil moisture datasets (ESA-CCI, SMERGE). The remainder of this paper is organized as follows: Section 2 presents the materials and methods and in Section 3, Section 4 and Section 5 are the results, discussion, and conclusions.

## 2. Materials and Methods

The number of coccidioidomycosis cases in AZ and CA were obtained from the Arizona Department of Health Services and US Centers for Disease Control, respectively. US Census Bureau American Community Survey was used to normalize annual coccidioidomycosis cases on the basis of annual population at the county scale.

The following two soil moisture datasets were used in this study: SSM was retrieved using ESA-CCI (version 3.3) [15] and RZSM from SMERGE (version 2.0) [22]. Note that bi-linear interpolation was used to fill in gaps that were present in the ESA-CCI dataset. ESA-CCI has a 0.25-degree spatial resolution and provides a daily estimate of surface soil moisture spanning the world between 1979 to the present. SMERGE was developed through the data fusion of ESA-CCI and Noah land surface model output generated by the North American Land Data Assimilation System, Phase 2 (version 2.8; NLDAS-2, NOAH0125_H.002; DOI 10.5067/47Z13FNQODKV) [23]. See [16] for a detailed description for the data fusion techniques used to form SMERGE. SMERGE is comprised of a climatology component (Noah) and an anomaly component that was derived based on the optimal weight between Noah and ESA-CCI. SMERGE spans CONUS with a 0.125-degree spatial resolution. This product provides a daily estimate of RZSM over four decades (1979 to 2019). For both ESA-CCI and SMERGE the average annual soil moisture value was calculated for each county using zonal statistics.

National Weather Service (NWS), Climate Prediction Center-generated ENSO indices which were used in this study for 2000 to 2016. Warm (El Nino) and cold (La Nina) episodes were defined using a threshold of ±0.5 °C (Oceanic Niño Index), which is based on a three-month running mean of Extended Reconstructed Sea Surface Temperature (version 2) sea surface temperature anomalies from the Niño 3.4 region (5° N–5° S, 120°–170° W).

Normalization of coccidioidomycosis cases, on a per 1000 persons basis, was necessary to account for annual changes in county population. This study followed the normalization procedures outlined in Coopersmith et al. [10]. Statewide, population growth in CA was 14% and for AZ 36% during the study period (2001 to 2018). Note that between 2001 to 2010, linear interpolation was needed to generate county population estimates. Average annual normalized cases varied across both AZ and CA and four categories of case occurrence were identified (Figure 1) including none; low (0 to 0.10 average cases per year); moderate (0.10 to 1.00 average cases per year); and high (>1.00 average cases per year).

Three approaches were used to evaluate case time series. In both AZ and CA, raw normalized counts (raw cases) and detrended counts (detrended cases) following the approach of [10] were analyzed. In AZ, there was an overcount of cases between 2009 to 2012 due to anomalous lab reporting. Comrie [9] addressed this discrepancy by dividing the case numbers during these years by two (referred to as half cases).

We used wavelet transform coherence (WTC) [24] to elucidate the inter-annual variability or cyclicity between ENSO and coccidioidomycosis cases/soil moisture products between 2002 to 2017. Wavelet coherence is a bi-variate framework that is useful for exploring interactions between time series within a continuous time and frequency space [25]. Prior studies that have used WTC to document teleconnections between ENSO and soil moisture include [26,27], which also provide more information about this methodology. WTC was applied on a county basis. Periodicities of 2.1, 3.0, 4.0, 5.0, 5.9, and 7.0 years were analyzed that correspond to the ENSO phenomena, and a Spearman’s rank correlation coefficient was utilized to derive a rank correlation (r^2^ value). This value represents the scale-averaged wavelet power between ENSO and coccidioidomycosis cases/soil moisture products. A r^2^ value of 0.5 was set as the minimum value at which phase relationships can be discerned with confidence [28]. A 12-month lag was applied between the ENSO and soil moisture consistent with the fact that hydrologic response can lag up to 12 months behind atmospheric forcing [27,29]. An additional 12 months (total 24 months) of lag was applied between ENSO and coccidioidomycosis cases, which has been noted in prior studies [7,11]. An analysis was conducted by dividing the study period into three eras (early 2002 to 2009; middle-2010 to 2012; late-2013 to 2017). For periodicities greater than 2.1 years, some truncation from the initial and final eras occurred to remove years outside of the cone of influence, which varies depending upon periodicity (see white dashed line on Figure 2). Start and end dates are indicated for early and late eras, respectively, in Table 1. Another aspect of WTC is its ability to discern phase behavior. In-phase (right arrow), anti-phase (left arrow), and whether a variable leads or lags in relation to ENSO were also identified (up and down, respectively).

## 3. Results

### 3.1. Coccidioidomycosis Cases

There were two clusters of moderate-to-high coccidioidomycosis cases in the study area, including the entire state of AZ and central CA (Figure 1). Counties with high case numbers (average annual normalized cases > 1.000 per 1000 persons) in AZ include Maricopa, Pima, and Pinal. In addition, eleven of the fifteen AZ counties demonstrated a significant upward trend in case numbers between 2001 to 2018. In central CA, Kern and Kings counties recorded high case numbers. In the nine central CA counties, with moderate-to-high cases, seven experienced a significant increase in cases from 2001 to 2018.

Figure 3 focuses on annual coccidioidomycosis case trends from Maricopa and Kern counties. In Maricopa County, three approaches were used to evaluate case trends (Figure 3a). Raw counts and detrended values [10] were consistent with each other on an inter-annual basis. For half cases [9], the mode at 2011 was muted, but still present. In Kern County, raw and detrended cases values exhibit very similar inter-annual variability (Figure 3b) with a bi-modal distribution. Modes were centered at 2011 and 2018.

The WTC analysis elucidates the relationship between ENSO and coccidioidomycosis cases. In AZ, higher r^2^ values were noted mostly for lower periodicities (2.1 and 3.0 years; Table 2). Statewide, only at the 2.1 year periodicity was a significant connection (r^2^ value > 0.5) noted between ENSO and coccidioidomycosis cases (raw and detrended) for only the first two eras (2002 to 2009; 2010 to 2012). Using half case estimates, no significant connection between ENSO and cases was discerned during any era statewide. Similar trends were observed when focusing on ENSO and coccidioidomycosis cases within counties that have high case counts. In general, r^2^ values were higher for an individual county compared with statewide averages. In terms of phase relationships, Maricopa county, which is representative, exhibited a generally in-phase relationship between ENSO and raw and detrended cases at a 2.1-to-3-year periodicity from 2004 to 2016 (Figure 2a,b). When examining half case numbers in Maricopa county cases lead ENSO by close to 90 degrees (Figure 2c).

The relationship between ENSO and coccidioidomycosis cases in central CA was different (Table 3). The average r^2^ of the nine counties in CA that had moderate-to-high case counts exhibited significant values during the first two eras (2002 to 2009; 2010 to 2012) at the 2.1 year periodicity. Conversely, the final era (2013 to 2014) had r^2^ values > 0.5 for periodicities of 5.9 and 7.0 years. In Kings County, CA, only during the early era (2002 to 2009) for the 2.1 year periodicity were r^2^ values > 0.5. Conversely, in Kern County, CA, USA, significant r^2^ values were recorded for all periodicities except for 4.0 years. In general, Kern County values were higher than the regional average. In terms of phase behavior, in Kern County, both raw and detrended cases exhibited nearly identical results with anti-phase behavior in the 2-to-3-year periodicities and generally in-phase results for the 5-to-7-year periodicities (Figure 4a,b).

### 3.2. Soil Moisture

The relationship between annual county soil moisture values for ESA-CCI and SMERGE is illustrated in Figure 5. Since SMERGE is in part derived from ESA_CCI, it is no surprise that these two datasets were highly correlated (r = 0.926). In Maricopa, Pima, and Pinal counties (AZ) both ESA-CCI and SMERGE exhibited a strong in-phase relationship with ENSO at a 7-year periodicity. These counties at the 2.1-to-3.0 periodicities demonstrated ESA-CCI-leading ENSO by close to 90 degrees (Figure 6a). In Kern and Kings counties (CA), both products experienced strong 2.1-to-3.0 and 5.9-to-7-year periodicities after 2009, which were generally in-phase for both ESA-CCI and SMERGE (Figure 6b).

In AZ, only the 2.1 and 7.0 year periodicities yielded significant r^2^ values (Table 4). SSM ESA-CCI displayed a stronger relationship with ENSO than RZSM SMERGE. At the 2.1 year periodicity, a significant connection (r^2^ value > 0.5) was noted statewide and in all counties that have high case counts for only the first two eras (2002 to 2009; 2010 to 2012). For the 7.0 year periodicity, up to a third of the counties had significant r^2^ values, including Maricopa County.

In the central CA counties, a broader spectrum of significant periodicities were noted for ESA-CCI and SMERGE (Table 5). The only periodicity lacking a significant relationship with ENSO was 4.0 years. At the 2.1-year periodicity, across the nine-county central CA region, only during the 2010 to 2012 era were significant r^2^ values recorded for both ESA-CCI and SMERGE. For Kern and Kings counties, very high r^2^ values (>0.8) for ESA-CCI were noted for the last two eras (2010 to 2012; 2013 to 2017). Likewise at the 3.0 year periodicity, high (r^2^ > 0.7) values were recorded for ESA-CCI in Kern and Kings counties. Significant r^2^ values (>0.5) at the county and regional levels for both ESA-CCI and SMERGE during the last two eras (2010 to 2012; 2013 to 2017) were present at the 5.0 year periodicity. Similar results were noted for the 5.9-and-7.0 periodicities with SMERGE yielding significant results also during the early era (2005 to 2009).

## 4. Discussion

### 4.1. Modulation of SSM and RZSM by ENSO

As indicated in Figure 5, there was a high correlation between SSM (ESA-CCI) and RZSM (SMERGE), which is expected given that SMERGE is in part derived from ESA-CCI. Both SSM and RZSM can impact the potential spread of coccidioidomycosis. The fungal habitat associated with coccidioidomycosis extends into the root zone [10]. Maddy [30] suggested that *Coccidioides* migrate into deeper soil in the hotter months and back to the surface during cooler conditions. Drying of SSM can facilitate the conditions that are conducive for coccidioidomycosis dispersion, i.e., the “grow and blow hypothesis” [1,10].

In AZ, mainly SSM from ESA-CCI was modulated by ENSO at the 2.1 and 7.0 year periodicities (Table 4). In this drier setting, excessive evaporation of summertime monsoon precipitation limits moisture infiltration into the deeper RZSM. Therefore, ENSO’s signal was confined to the surface. At the 2.1 year periodicity, higher r^2^ values were noted for the 2010 to 2012 period. This was a period when ENSO was in-phase with the Pacific Decadal Oscillation (PDO), which reinforces the strength of the ENSO signal [31,32,33]. For the 7.0 year periodicity from Maricopa, Pima, and Pinal counties (AZ) an in-phase relationship was noted with ENSO.

Central CA has a different hydroclimate regime. Precipitation is concentrated in the winter months when temperatures are lower. Therefore, rainfall penetrates deeper into the soil profile. Therefore, in central CA, both SSM and RZSM exhibit a strong connection with ENSO (Table 5). Both low (2.1 to 3.0 years) and high (5.0 to 7.0 years) periodicities were embedded within SSM from ESA-CCI. However, the deeper RZSM depicted by SMERGE has a stronger signal than ESA-CCI for the higher periodicities (5.0 to 7.0 years). Barco et al. [34] records a similar tendency when examining the connection between ENSO and groundwater levels in CA. Deeper wells tended to record higher periodicities as opposed to shallower wells. In conclusion, there is ample evidence that ENSO modulates soil-moisture interannual variability in both AZ and central CA.

### 4.2. Connection between ENSO and Coccidioidomycosis Cases

The connection between ENSO and coccidioidomycosis cases has greater complexity. In AZ, raw counts and detrended case values have a strong relationship with ENSO at the 2.1 year periodicity in the three counties defining the core endemic zone (Table 2). Using the approach of Comrie [9], cutting case numbers in half between 2009 to 2012 (half cases), minimizes the strength of this connection. ENSO and coccidioidomycosis cases based on raw counts and detrended have r^2^ values that exceed those recorded for ENSO and SSM ESA-CCI. Due to a host of non-climatic factors (described in Section 4.4), the expectation would be that the connection between ENSO and cases would be less than ENSO and SSM ESA-CCI. The half cases best match the above supposition. In fact, the half cases lead ENSO by a quarter cycle (≅0.5 year). This matches the shorter lag times noted in AZ for the onset of coccidioidomycosis cases [6] and the phase behavior between ENSO and SSM ESA-CCI (Figure 6a). Raw counts and detrended time series are quite similar in AZ (Figure 3a). So, the historical reporting issues likely inflated the spike in cases between 2009 and 2012 and the apparent strong connection with ENSO is artifactual. In short, the inflation in case numbers occurs during the same period as the intensification of ENSO by PDO, generating the very high r^2^ values between ENSO and cases in Table 2 for raw and detrended cases.

In central CA, apparent connections between ENSO and coccidioidomycosis cases were even more pronounced. Both short (2.1 to 3.0 years) and long (5.0 to 7.0 years) periodicities were discerned. An anti-phase relationship between ENSO and coccidioidomycosis cases was noted for the 2.1 to 3.0 years periodicities, indicating that ENSO and cases were inversely related. The best case for establishing a connection between ENSO and coccidioidomycosis cases in central CA was during the final era (2013 to 2014) for the higher periodicities (5.0 to 7.0 years). Both SSM ESA-CCI and RZSM SMERGE have robust r^2^ values with significant but lower r^2^ values for ENSO and cases recorded (Table 3 and Table 5). It should be noted that both Kern and Kings counties in the central CA region used a lab-based system for case reporting before 2010, resulting in a more accurate time series [7], unlike CA as a whole.

### 4.3. Issues with Coddidiodomycosis Case Numbers

There were two main limitations to the approaches applied in this work that include discrepancies associated with case reporting over time and the confounding effects of the travel of residents in and out of the endemic region. Issues in AZ were associated with a change in the laboratory definition that approximately doubled case numbers between 2009 to 2012 [35]. Although coincident increases in Valley fever incidence and hospitalization rates around 2009 suggests that this peak in cases cannot be attributed to a change in diagnosis alone [6]. In CA, Zender and Talamantes [36] indicates that some of the spike recorded around 2011 might be artifactual in nature. However, ref. [2] asserts that the interannual variability in cases is indicative of changes in disease incidence to some extent. The coarse annual resolution of this study has the benefit of generalizing errors present in finer scale reporting. For example, at the monthly time scale errors result from delays in reporting and the onset of disease. A final consideration is that the reporting jurisdiction may include some cases from non-residents adding noise to the case counts not attributable to any local factors.

### 4.4. Connection between Non-Climatic Factors and Coccidioidomycosis Cases

Non-climatic factors may also contribute to the rate of disease. This result directly supports the conclusion of [4,7] that only a moderately positive significant association is evident between coccidioidomycosis cases and climate variability. Specifically, in central CA, it has been suggested that the cause for coccidioidomycosis cases were more related to anthropogenic impacts on the landscape, such as agricultural activity, soil disturbances associated with construction and development, and archeology activity, which are unconnected to ENSO [37]. Such disturbances can aerosolize spores. For example, cropland area has as strong positive correlation with coccidioidomycosis cases [6]. Fisher et al. [38] identified a mismatch between the endemic zone for coccidioidomycosis that is in the order of hundreds of kilometers versus growth sites that are located at the scale of centimeters to meters. The county level data used in this study were bounded by these two extremes on spatial scales. Pianalto [39] found a higher incidences of coccidioidomycosis around the periphery of Tuscon, AZ as opposed to the city center based on the analysis of cases within zip code areas. These peripheral areas were subjected to more landscape disturbance that could facilitate the dispersal of fungal spores. Taylor and Barker [40] link the disturbance of soil associated with burrowing rodents with the spread of fungal spores.

Another non-climatic variable that could influence these results is a change in population demographics. Individuals who are at the greatest risk for coccidioidomycosis include those who are immunocompromised as well as very old persons. Additionally, major non-white ethnic groups (African American and Filipino) are more likely to contract severe Valley fever compared with Whites [41,42]. In addition, persons with outdoor occupations involved with disturbing soils have a higher likelihood of developing Valley fever [43]. All these non-climatic factors, while not dominant, can add noise to an already noisy dataset.

## 5. Conclusions

In conclusion, this study applied the novel approach of WTC to see if there was a link between ENSO and coccidioidomycosis cases. While there is a strong connection between ENSO and soil moisture, ENSO’s possible control on coccidioidomycosis is more difficult to elucidate. In AZ, the approach of dividing case numbers by two between 2009 to 2012 is best justified by supporting information. Only SSM and not RZSM was modulated by ENSO. In Maricopa and Pinal counties, ENSO exerts a moderate but significant control on cases at a short (2.1 year) periodicity. In central CA, both SSM and RZSM were impacted by ENSO. ENSO and cases exhibit significant periodicities at a longer timescale (5-to-7 year). These results support the premise that ENSO, and hence soil moisture, has an inter-annual variability that can be connected with the occurrence of coccidioidomycosis cases. Therefore, this study is supportive of the “grow and blow” hypothesis. The approach described here can be expanded both spatially and temporally. With climate change, the endemic region for coccidioidomycosis is predicted to expand to cover most of the western United States. Likewise, with a lengthened time series, the role of climate modulation resulting from PDO could be better discerned.

## Figures and Tables

**Figure 1 ijerph-19-07262-f001:**
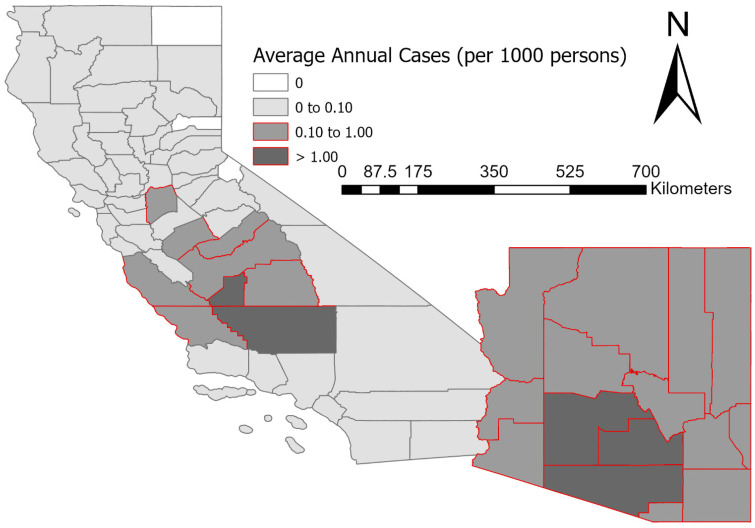
Average annual (2001 to 2018) normalized coccidioidomycosis cases. Cases were classified into four categories including none; low (0 to 0.10 average cases per year); moderate (0.10 to 1.00 average cases per year); and high (>1.00 average cases per year).

**Figure 2 ijerph-19-07262-f002:**
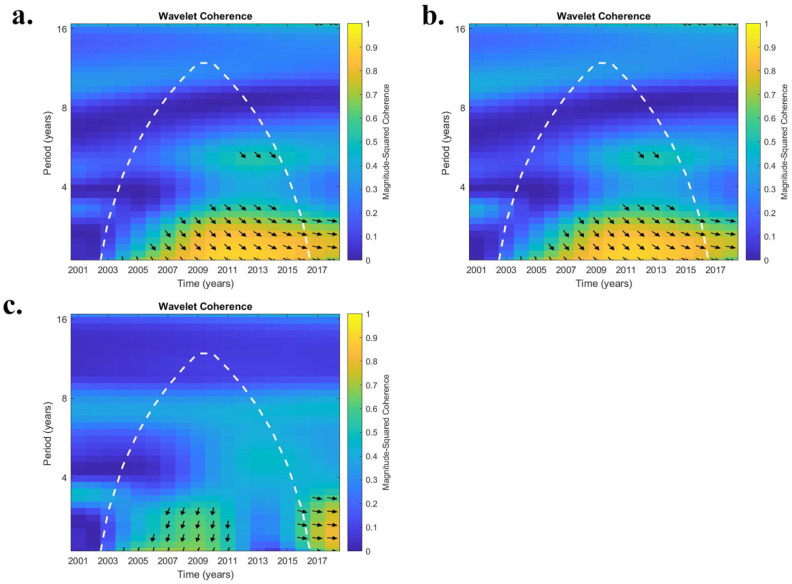
Wavelet transform coherence (WTC) analysis between El Nino Southern Oscillation (ENSO) and coccidioidomycosis cases from 2002 to 2017 from Maricopa County, AZ. (**a**) represents raw cases, (**b**) detrended cases, and (**c**) half cases that were divided by two between 2009 to 2012. The white dashed line indicates the cone of influence (Table 1), with only results inside of the cone considered valid. Arrows indicate the relative phase of the relationship (right arrows, in-phase; left arrows, anti-phase; down arrows, coccidioidomycosis cases lead ENSO by 90 degrees; up arrows, ENSO lead coccidioidomycosis cases by 90 degrees).

**Figure 3 ijerph-19-07262-f003:**
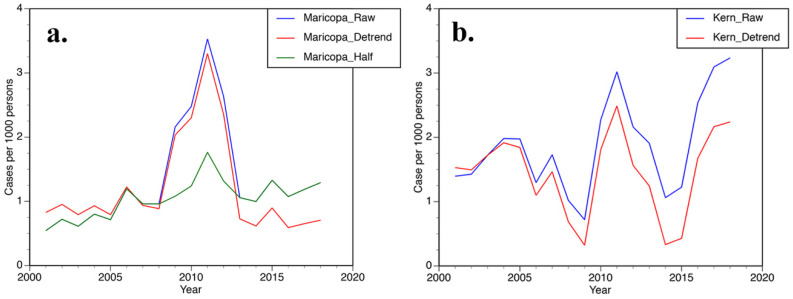
Annual trend in coccidioidomycosis cases from (**a**) Maricopa County, Arizona (AZ) and (**b**) Kern County, California (CA). Raw and detrended cases are presented. Half indicates cases divided by two between 2009 to 2012.

**Figure 4 ijerph-19-07262-f004:**
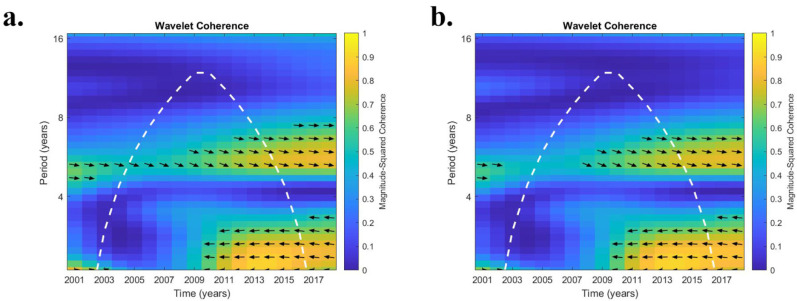
Wavelet transform coherence (WTC) analysis between El Nino Southern Oscillation (ENSO) and coccidioidomycosis cases from 2002 to 2017 from Kern County, CA, USA. (**a**) represents raw cases, (**b**) detrended cases. All else as indicated in Figure 2.

**Figure 5 ijerph-19-07262-f005:**
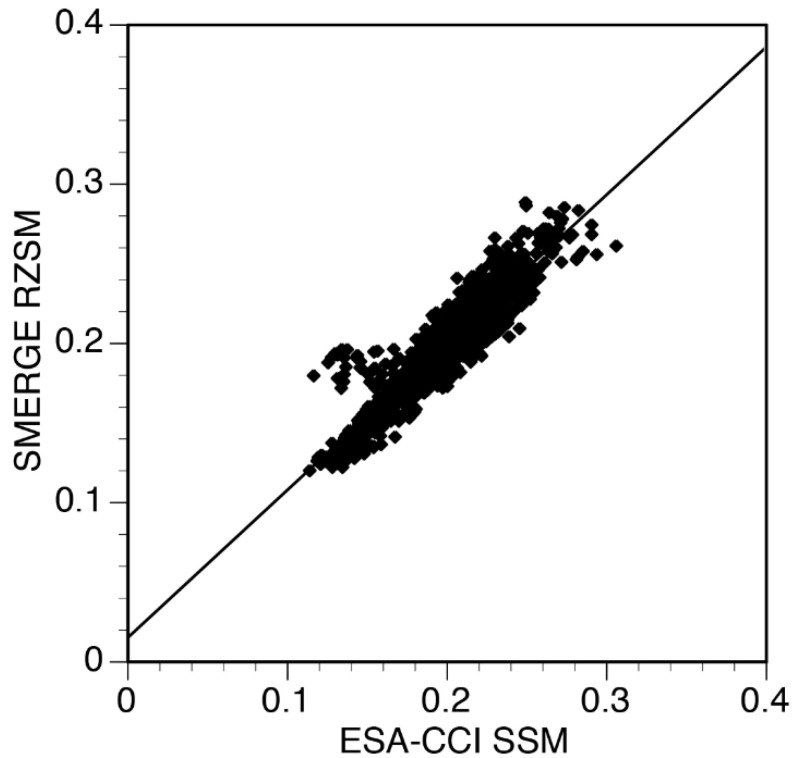
A strong linear relationship (r = 0.926) is seen between SSM from ESA-CCI and root zone soil moisture (RZSM) from Soil MERGE (SMERGE).

**Figure 6 ijerph-19-07262-f006:**
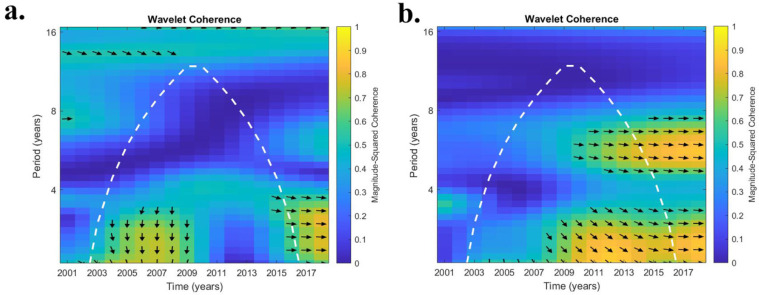
Wavelet transform coherence (WTC) analysis between El Nino Southern Oscillation (ENSO) and surface soil moisture (SSM) from European Space Agency Climate Change Initiative (ESA-CCI) from 2002 to 2017. (**a**) Pinal County, AZ, (**b**) Kern County, CA, USA. All else as indicated in Figure 2.

**Table 1 ijerph-19-07262-t001:** Start date for early era and end date for late era.

Periodicity (Years)	Early Era (Start)	Late Era (End)
2.1	2002	2017
3.0	2003	2016
4.0	2004	2015
5.0	2005	2014
5.9	2005	2014
7.0	2006	2013

**Table 2 ijerph-19-07262-t002:** Average r^2^ values for ENSO and coccidioidomycosis cases in Arizona.

		Raw Cases			Detrended Cases			Half Cases		
Location	Perd. (yrs)	Start to 2009	2010 to 2012	2013 to End	Start to 2009	2010 to 2012	2013 to End	Start to 2009	2010 to 2012	2013 to End
Maricopa	2.1	0.587	0.897	0.832	0.606	0.897	0.814	0.525	0.516	
Pima	2.1	0.778	0.704	0.533	0.793	0.702	0.509			
Pinal	2.1	0.780	0.871	0.775	0.786	0.870	0.748	0.674		
**Avg AZ**	**2.1**	**0.502**	**0.543**			**0.542**				
Maricopa	3.0		0.709	0.666		0.707	0.653			
Pima	3.0		0.648	0.678		0.644	0.660			
Pinal	3.0	0.569	0.762	0.736	0.549	0.759	0.717	0.566		
**Avg AZ**	**3.0**	**None**								
Pinal	4.0		0.523			0.503				
**Avg AZ**	**4.0**	**None**								
Maricopa	5.0			0.519			0.507			
Pinal	5.0		0.519							
**Avg AZ**	**5.0**	**None**								
**All**	**5.9**	**None**								
**All**	**7.0**	**None**								

Statewide averages in bold. Only r^2^ values greater than 0.5 are indicated.

**Table 3 ijerph-19-07262-t003:** Average r^2^ values for ENSO and coccidioidomycosis cases in central California.

		Raw Cases			Detrended Cases		
Location	Perd. (yrs)	Start to 2009	2010 to 2012	2013 to End	Start to 2009	2010 to 2012	2013 to End
Kern	2.1		0.746	0.831		0.745	0.857
Kings	2.1	0.554			0.608		
**Avg Central CA**	**2.1**	**0.588**	**0.659**			**0.514**	
Kern	3.0		0.543	0.646		0.542	0.649
**Avg Central CA**	**3.0**	**None**					
**All**	**4.0**	**None**					
Kern	5.0		0.561	0.624		0.551	0.616
**Avg Central CA**	**5.0**	**None**					
Kern	5.9		0.625	0.714		0.603	0.698
**Avg Central CA**	**5.9**			**0.515**			**0.529**
Kern	7.0			0.504			
**Avg Central CA**	**7.0**			**0.532**			**0.505**

Regional averages in bold. Only r^2^ values greater than 0.5 are indicated.

**Table 4 ijerph-19-07262-t004:** Average R^2^ values for ENSO and soil moisture in Arizona.

		SSM ESA-CCI			RZSM SMERGE		
Location	Perd. (yrs)	Start to 2009	2010 to 2012	2013 to End	Start to 2009	2010 to 2012	2013 to End
Maricopa	2.1	0.516	0.656				
Pima	2.1	0.683	0.719				
Pinal	2.1	0.699	0.765				
**Avg AZ**	**2.1**	**0.582**	**0.751**				
**Avg AZ**	**3.0**	**None**					
**Avg AZ**	**4.0**	**None**					
**Avg AZ**	**5.0**	**None**					
**Avg AZ**	**5.9**	**None**					
Maricopa	7.0	0.765	0.720	0.706	0.574	0.505	
Pima	7.0	0.539					
Pinal	7.0			0.522			
**Avg AZ**	**7.0**	**None**					

Statewide averages in bold. Only r^2^ values greater than 0.5 are indicated.

**Table 5 ijerph-19-07262-t005:** Average R^2^ values for ENSO and soil moisture in central California.

		SSM ESA-CCI			RZSM SMERGE		
Location	Perd. (yrs)	Start to 2009	2010 to 2012	2013 to End	Start to 2009	2010 to 2012	2013 to End
Kern	2.1	0.520	0.848	0.767			
Kings	2.1	0.543	0.968	0.900		0.641	
**Avg Central CA**	**2.1**		**0.591**			**0.600**	
Kern	3.0		0.683	0.650			
Kings	3.0		0.748	0.789			
**Avg Central CA**	**3.0**	**None**					
**Avg Central CA**	**4.0**	**None**					
Kern	5.0		0.548	0.698		0.564	0.664
Kings	5.0		0.573	0.732		0.549	0.677
**Avg Central CA**	**5.0**			**0.633**		**0.508**	**0.611**
Kern	5.9		0.608	0.746	0.610	0.739	0.820
Kings	5.9		0.695	0.831	0.621	0.769	0.856
**Avg Central CA**	**5.9**		**0.605**	**0.756**	**0.636**	**0.755**	**0.836**
Kern	7.0			0.554	0.518	0.584	0.636
Kings	7.0	0.524	0.610	0.665	0.582	0.631	0.675
**Avg Central CA**	**7.0**		**0.548**	**0.614**	**0.589**	**0.643**	**0.690**

Regional averages in bold. Only r^2^ values greater than 0.5 are indicated.

## Data Availability

All data used in this study are publicly available and include: Arizona Department of Health Services (https://www.azdhs.gov/preparedness/epidemiology-disease-control/index.php#data-stats-past-years); US Centers for Disease Control (https:www/cdc.gov/fungal/diseases/coccidioidomycosis/statistics.html); US Census Bureau American Community Survey (https://www.census.gov/programs-surveys/acs/data.html); ESA CCI SSM (https://data.ceda.ac.uk/neodc/esacci/soil_moisture/data/daily_files/COMBINED/v03.3/); and SMERGE RZSM (https://disc.gsfc.nasa.gov/datasets/SMERGE_RZSM0_40CM_2.0/summary).

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
