# Peer review of "Coccidioidomycosis (Valley Fever), Soil Moisture, and El Nino Southern Oscillation in California and Arizona"

_ijerph, 2022, doi:10.3390/ijerph19127262_

Round 1

Reviewer 1 Report

This manuscript is enormously improved over original draft.  There is significant information on coccidioidomycosis included as well as tying the current work to what is known in the field already.  Very good job of aligning this data to contribute to the field overall.

Please address the following minor typographical and grammatical issues:

Line 31 - elderly

line 46-47 - sentence fragment - fix

line 118 - typo anomalous

line 139-140 - correct sentence

line 162 - remove AZ - defined in previous sentence

line 183 -  insert comma between county and cases

line 211 - space between up and to

line 222 - add dash (-) between nine and coutny to improve readability

line 245 - typo - "migrates"

line 282 - remove "of"

line 298 - consider add "for case reporting" after "lab-based sysem"

line 302 - "effects" not "affects" (verb)

line 305-306 - not a complete sentence - correct

line 311 typo - time scale twice

line 330 - link, not links

line 336 - remove "case"

line 337 - suggest "likelihood" or probability not susceptibility, which is more of a personal health than an epidemiologic term

line 352 - needs a comma

Author Response

I have made all the revisions requested by reviewer 1. Thank you

Reviewer 2 Report

All reviewer comments have been addressed accordingly and the manuscript is in a good shape for publication.

Author Response

Thank you

Reviewer 3 Report

Coccidioidomycosis (Valley fever), Soil Moisture, and El Nino 2 Southern Oscillation in California and Arizona

Reviewer Comments

This revision is quite well executed. The authors have in their revisions satisfied the chief concerns I had in my initial review. The paper is now quite well-balanced in terms of results and limitations. We found it difficult to tease out environmental vs. demographic and urbanization factors, which the authors acknowledge. I believe this paper contributes new analytical methods that, going forward, will continue to clarify what we have termed the ‘mystery bug of the Universe.’ This being said, I had only a few editorial comments, below. I would appreciate the authors explaining their method for splitting the study period in 3 eras, since it raises the haunting spectre of ‘binning bias’ seen typically in producing histograms. I believe that, with these very minor revisions, the paper is good to go.

Abstract: A statistical analysis identified two areas IN Arizona and central California, with 12 a moderate-to-high number of coccidioidomycosis cases.

Line 46 fragment: Although [9] showed no consistent connection between dust 46 storms and Valley fever.

Line 65: summer Southwestern Monsoon (July to September). Should it be termed the North American Monsoon? I’ve never seen ‘Southwestern Monsoon.’

Line 135: Analysis was conducted by dividing the study period into three eras (2002 to 2009, 2010 to 2012, 2013 to 2017). Please provide brief rationale for these splits.

Author Response

I have made all the revisions requested by reviewer 3. New changes are indicated in blue.

My answer to comment 4 from reviewer 3 is as follows. I split the data into three eras for clarity. In the original paper I included data from all years, and I believe there was some confusion in interpreting the results. I choose 2010 to 2012 as a period when both there was intense ENSO

This manuscript is a resubmission of an earlier submission. The following is a list of the peer review reports and author responses from that submission.

Round 1

Reviewer 1 Report

This is a paper in which the authors assess several decades of soil moisture data and ENSO data for their relationship to rates of coccidioidomycosis in CA and AZ.  The work is extensive.  The manuscript is difficult to understand unless the reader is a soil or climate scientist.  I particularly did not understand the figures, but presume this is due soil/climate not being my field.  I had a hard time getting from the results to the conclusions in the discussion.  The discussion sections were not clear to me in leading up to the conclusions in the last paragraph, and I don’t understand the ENSO relationship to cocci in CA or AZ from reading this.  Data on the annual or decadal coccidioidomycosis rates (or both if there are times on which the comparisons focus), would help the reader understand the underpinnings of this research. There are graphs of the annual cases for both AZ and CA on the states’ respective websites, and there are some combined graphs on the CDC website.  The manuscript requires significant editorial attention to word choice, typographical errors and making sure the sentence structure and grammatical features are correct when revising. 

Coccidioidomycosis is misspelled in the title of the article.

Line 7 – coccidioidomycosis should not be capitalized or italicized

Line 8 – coccidioidomycosis is an infection, not an affliction, technically

Line 18 – somewhere in this manuscript, you have to define what you mean by an outbreak, because that is an acute increase in an endemic disease that comes and goes, or it is an appearance of a disease suddenly that was not there before.  Coccidioides has a strong endemic presence in the areas where you are analyzing it. See CDC webpage for definition, examples, and references for coccidioidomycosis outbreaks: https://www.cdc.gov/fungal/diseases/coccidioidomycosis/statistics.html

Line 28 –The entire population of the endemic area does not get infected with Coccidioides, only about 3% of people per year in the heavily endemic areas of AZ and less most other places.  Only 40% of infections are symptomatic, resulting in about 150,000 infections per year.

Line 31-  reference please

Line 33, 41, and wherever else  - coccidioidomycosis is lower case

Line 47 – fungal, not fungi; fungi” is the plural of “fungus”

Line 53 – word missing – ‘that”

Line  69 – I don’t understand “positive(negative)” – please clarify

Line 69 – which include what??  This sentence should probably be reconsidered and clarified entirely.

Line 77 – data is plural, not singular

Line 78, 90, everywhere else-  coccidioidomycosis in not cap or italics.  It is a disease.  Only the organism is italicized and capitalized

Line 101 – please state what this is and don’t just provide a reference number.  The reader does not know what you are alluding to.

Line 143 – include what?

Line 151 – word order

Line 159 – there is an extra “of”

Line 171 – “case”

Line 173 – which counties? You only named Kern

Line 175 – sentence fragment –please correct

Line 228-229 – I don’t understand this connection. Can you clarify why this is relevant to the results?

Also, what is the conclusion from the results of this data.  I am not exactly clear on the relationship between ENSO and soil moisture after I read this, and I don’t understand the relationship of these things to the different periodicities and how this matters to soil moisture and cocci.  I finally went and read one of your references (Coppersmith, et.al.) and that paper is much more explanatory of the “story” of the potential relationships of cocci to climatic/soil moisture factors.  Can you please clarify these relationships for this manuscript. This paper might benefit significantly from more visual information about the rate of disease over time, such as the table used in the Coppersmith paper, which gave the reader a visual on the disease rate in CA and AZ.  Even the graphs do not address the rate of disease in any way that gave me a clear picture of number of cocci cases, or changes in rate of disease in relationship to soil moisture or ENSO.

Line 256 – fungal

Line 258 – this needs a reference. Also, in infectious disease terms, coccidioidomycosis is not a vector-borne disease, which has a very specific meaning, so suggest choosing a different term here.

Line 259-260 – I don’t understand this relationship between stronger signals related to ESA-CCA vs cocci cases

Line 263 – not casual, causal.  Also, this article does not address coccidioidomycosis itself hardly at all, including no showing of the data on cases over the years, especially the last 20 years, which is easily accessed on the CDC website and possibly in articles.  No discussion is made at all about the general trend upward in cocci cases in the last 10-15 years, especially in CA.    A spike in AZ cases in 2011 was partially the result of a change in reporting/case definition, which was modified again later.

Line 263 – “Outbreaks” of coccidioidomycosis are not defined or discussed as to what an “outbreak” is in the context of endemic disease.

Starting Line 320 – If I understand what you wrote in the results, there is a difference between Arizona and California and the relationship between the periods of ENSO.  This does not seem to discuss these differences or take them into account in understanding the soil moisture/disease rate relationship.  Maybe I missed something.

Line 331 – temperature does not seem likely to have a direct impact on dispersion of spores, unless you have a reference for this.

Line 336 – extend is used incorrectly

Line 340 – this needs a reference.  In the onset of individual disease, “climate” did not expose anyone to the spores. It may have changed conditions that affected numbers of spores, or dispersal of spores, as you have stated earlier.  Consider reword this more accurately. Also, coccidioidomycosis does not disappear from the endemic areas regardless of the ENSO conditions, so I do not think you said what you intended to say here.  See Pappagianis on the Northridge earthquake to look at a case of a true environmental event that led to cases in places they do not occur, and then they did not occur again there.

344 – typo.  Also, “onset” of disease is not correct.  “Rate” would be a better word since this paper is not talking about dates of diagnosis, but lumped data regarding reported cases annually.  This word is used incorrectly elsewhere in this manuscript as well.  Please review and correct.

360 - typo

Line 367 – primary, not primal, which means more or less instinctive. 

Reviewer 2 Report

ijerph-1491858

Coccidiodomycosis (Valley Fever), Soil Moisture, and El Nino 2 Southern Oscillation in California and Arizona.

In this report, the El Nino Southern Oscillation (ENSO) effect on two different soil moisture surrogates, soil surface moisture and Root Zone Soil Measure (RZSM), and the correlation of these figures with case rates of coccidioidomycosis (CM) in 74 counties of Arizona and California.  The authors find that SSI correlates better with CM case rates than did RZSM and that ENSO had different effects on moisture characteristics in California compared to Arizona.

Main comments

I find this manuscript tantalizing in what relationships it proposes exist.  My large problem with this presentation is that most of the results appear to focus upon the effect of ENSO on the two different estimates of soil moisture and do not clearly present either in figures or tables the relation ship to CM case rates. If the authors disagree with this assessment, it underscores how poorly what they purport to demonstrate actually could be comprehended by this reviewer. 

One problem I have with the approach is that many of the counties that are analyzed have little or no endemic Coccidioides.  Case rates in those areas are influenced by the residents’ travels to endemic regions and it is impossible to know the exact geographic source of their exposure.  To my mind, it’s perfectly reasonable to look at the effect of ENSO on soil moisture but it is meaningless to consider the relationship of soil moisture to case rates in those non-endemic parts of California or Arizona.

Another problem I have with the California analysis is that case rates have changed considerably over time as there has been increased interest in CM in the state and in some counties case reporting has become more efficient.  This could be significantly reduced in not eliminated if the California analysis was limited to Kern County or to Kern, Kings, and San Louis Obispo which have been more consistent in reporting.

Other points

Coccidioidomycosis is misspelled in the title.

Capitalization of “Valley Fever” should be either “Valley fever” (CDC usage) or “valley fever” (Associated Press usage).

Coccidioidomycosis should not be italicized.  Coccidioides is but the disease-name is not.

Line 31:                This disease is not more prevalent in the very young.

Line 49:                the Stacey reference is ref 27 and needs to be recompiled.

Lines 50-52:        Do these references relate moisture to coccidioidomycosis?  They don’t seem to.

Line 59:                Define CONUS.

Lines 159-160:    What is problematic about this description is that much of the land in the 11 counties referred to have virtually no evidence for endemic Coccidioides

Table 1:                I guess the numbers in the first column are a periodicity constant used in the calculations? If so, this needs to be shown in the table or described in the legend.

Lines 168-169:    Graham and Yavapai Counties are pretty much non-endemic.

Reviewer 3 Report

The manuscript is potentially a good contribution however it  needs to be major revisions  concerns data treatment and methods: go from simple time series illustration methods then employ wavelet transformation. 

Reviewer 4 Report

Coccidiodomycosis (Valley Fever), Soil Moisture, and El Nino 2 Southern Oscillation in California and Arizona

Reviewer Comments

Summary: This paper presents results from two decades of synoptic climate data, assessing the teleconnection between Valley Fever and ENSO events during this period. The paper is well written and pretty well organized. To my knowledge, this paper is among the first to associate Cocci with synoptic climate data, thus is novel and could attract reader interest. (Most prior work focused on remote-sensing proxies.) My chief concern is the relatively broad scale of climate data may provide neither the model accuracy, nor the map scale, useful to medical authorities needing to know the ‘when and where’ of outbreaks. I will in the list following suggest revisions I believe will clarify the chief thrusts of this paper.

  1. Abstract: Would it make sense to add a brief sentence clarifying why 2009-2012 produced strongest correlations?
  2. Should there be a short para following line 47 that reveals to the reader how the paper is organized? I think it would be informative.
  3. Not sure if the reference to Stacy’s 2012 work should cite him directly, or be a numbered reference, like the others?
  4. At line 50, the ‘sentence’ beginning with ‘Or…’ seems to be a fragment.
  5. Got a reference for the Noah land surface model?
  6. Line 75: Mixing singular and plural. Authors should likely proofread entire paper a bit better.
  7. Line 79: Two soil moisture datasets were used in this study: (i.e., use a colon, not a period)

I have a concern about the ESA-CCI resolution: ESA CCI SM products have provided remotely-sensed surface soil moisture (SSM) content with the best spatial and temporal coverage thus far, although its output spatial resolution of 25 km is too coarse for many regional and local applications.

Author: Jovan Kovačević, Željko Cvijetinović, Nikola Stančić, Nenad Brodić, Dragan Mihajlović

Cited by: 5

Publish Year: 2020

  1. My concern, as implied above, is that most studies I know believe Coccidioides is limited to spatially focal (i.e., local) habitats, hence some prior work using zipcode-level incidence. This is a complicated good news-bad news situation for you, in my view: Good news is that broad county-level scales may cancel modelling errors; but the bad news is county-level results do little to inform medical authorities about which hospitals to staff, and when. Can you provide a sound rationale for using ESA-CCI SM given the apparent mismatch in spatial resolution between the ESA-CCI data and the scale of action of the fungus? Would the following paper thus be useful?

New Downscaling Approach Using ESA CCI SM Products for Obtaining High Resolution Surface Soil Moisture; Jovan Kovaˇcevi´c* , Željko Cvijetinovi´c, Nikola Stanˇci´c, Nenad Brodi´c and Dragan Mihajlovi´c Faculty of Civil Engineering, University of Belgrade, Bulevar kralja Aleksandra 73, 11000 Belgrade, Serbia; zeljkoc@grf.bg.ac.rs (Ž.C.); nstancic@grf.bg.ac.rs (N.S.); nbrodic@grf.bg.ac.rs (N.B.); draganm@grf.bg.ac.rs (D.M.) * Correspondence: jkovacevic@grf.bg.ac.rs Received: 18 February 2020; Accepted: 30 March 2020; Published: 1 April 2020

Line 92: Should the county-level scale of your model be stated in the introductory sentences of your Materials and Methods—or perhaps even earlier, in the Abstract and Introduction? And I must wonder if a county-level model, even if quite robust, gives the Cocci docs/medical infrastructure the data they need to fight Cocci (per comment above)? Using finer-grained data (e.g., ETM+) enables following the ecology of the fungus (e.g., read Maddy’s classic 1957 paper; but see also more contemporaneous work by Fred Fisher and/or Frederick Pianalto, et al.), including coincidence with rodent populations (hence Stacy’s focus on NDVI greenness). Some of the latest speculation by the Cocci Research Group, at their annual meeting, lends support to a ‘rat hole’ hypothesis, which favors modeling finescale rodent habitat to isolate Coccicioides.  In fact, you should check out Comrie’s 2021 paper in Geohealth, in which he argues the need to focus on soil disturbance and the role of burrowing animals.

  1. I wonder about the R values presented: An R value of 0.5 is only modestly significant, at best. Do you have a rationale for selecting a Spearman’s R of 0.5? What was the reason for not using OLS regression?
  2. Following up on #9 above, I need to ask whether county level data, smoothed using WTC and kriging, is the appropriate model scale to present? Medical authorities cannot use it readily. Given processing by both WTC and kriging may ‘oversmooth’ the data and remove compelling trends, I guess I’m not surprised by the modest R values.
  3. I concur with your discussion in lines 320-342, which introduces your thoughts on non-ENSO related triggers. It made me think about the relevance of demographics and urbanization: one possible explanation for the AZ results is that AZ populations tend to be old; and the ‘geezers’ are the folks most vulnerable to cocci. And land-use change is, as you’ve suggested, apparently an important driver of cases. We know, moreover, that medical authorities have been slow over the years to diagnose cocci, thinking it’s community-acquired pneumonia; and there was a period when we seemed to have an ‘epidemic of reporting’ where cases soared due to mandated reporting requirements. Bottom line: the cocci stats are inherently noisy; so I sympathize with folks trying to use them to build robust models.
  4. The Discussion at 4.1 is difficult to follow. There is no obvious analysis of the different results in AZ vs. CA.
  5. Section 4.3 contains some valuable insights; and I think cocci science needs a model that folds in non-climate variables, hence my suggestions you at least mention some of the classic papers on the ecology of Coccidioides, and also at least offer a critique of papers using non-climate proxies to model fungal habitats. That you have striven to model ‘the mystery bug of the Universe’ with minimal data requirements is commendable. But given very modest relationships between the predictor variables and response variable, it is all the more important to present a comprehensive review of prior non-climate-driven models.
  6. Your Conclusion definitely needs more heft: You should in the Conclusions summarize your findings, cite the limitations of the research, and recommend directions for future work. Citing recommendations may have the effect of staking your claim on future research.